# Topology and Function of the *S. cerevisiae* Autophagy Protein Atg15

**DOI:** 10.3390/cells12162056

**Published:** 2023-08-12

**Authors:** Lisa Marquardt, Marco Montino, Yvonne Mühe, Petra Schlotterhose, Michael Thumm

**Affiliations:** Institute of Cellular Biochemistry, University Medicine, Humboldtallee 23, D-37073 Goettingen, Germany

**Keywords:** autophagic body, lysis, macroautophagy, microautophagy, yeast, phospholipase

## Abstract

The putative phospholipase Atg15 is required for the intravacuolar lysis of autophagic bodies and MVB vesicles. Intracellular membrane lysis is a highly sophisticated mechanism that is not fully understood. The amino-terminal transmembrane domain of Atg15 contains the sorting signal for entry into the MVB pathway. By replacing this domain, we generated chimeras located in the cytosol, the vacuole membrane, and the lumen. The variants at the vacuole membrane and in the lumen were highly active. Together with the absence of Atg15 from the phagophore and autophagic bodies, this suggests that, within the vacuole, Atg15 can lyse vesicles where it is not embedded. In-depth topological analyses showed that Atg15 is a single membrane-spanning protein with the amino-terminus in the cytosol and the rest, including the active site motif, in the ER lumen. Remarkably, only membrane-embedded Atg15 variants affected growth when overexpressed. The growth defects depended on its active site serine 332, showing that it was linked to the enzymatic activity of Atg15. Interestingly, the growth defects were independent of vacuolar proteinase A and vacuolar acidification.

## 1. Introduction

Macroautophagy (hereafter autophagy) delivers superfluous or damaged cytosolic material, including parts of organelles, for degradation to the lysosome (vacuole). Autophagy and its molecular machinery are highly conserved from yeast to humans and have attracted great medical interest [1,2,3,4]. In *S. cerevisiae*, which was used in this project, autophagy starts with the de novo assembly of double-membraned phagophores (isolation membranes). They are elongated and closed to an autophagosome. Eventually, the outer membrane of an autophagosome fuses with the vacuole and an autophagic body (AB) still enclosed by a single membrane is released into the vacuole lumen. To allow the vacuolar hydrolases access to the content of ABs, their membrane must be lysed.

Membrane lysis is a unique and potentially dangerous feature of eukaryotic cells. We thus have a long-standing interest in the mechanism of this process since we and others identified Atg15 to be essential for the intravacuolar disintegration of autophagic bodies [5,6,7]. Atg15 is an integral membrane protein with a predicted transmembrane domain near its aminoterminus and contains a GXSXG lipase/esterase motif, whose serine 332 is essential for activity. Atg15 mainly reaches the vacuole via the MVB pathway. Defects in this pathway, especially when combined with a Atg15-HDEL variant, which is recycled back to the ER, suggest that Atg15 acts within the vacuole [5]. This is nicely underlined by a recent study, which showed that the amino-terminal transmembrane domain of Atg15 contained its sorting information for entry into the MVB pathway and that an Atg15 variant misdirected to the vacuolar membrane was biologically active [8]. Interestingly, Atg15 is also required for the degradation of micronucleophagic vesicles [9]. During micronucleophagy (also termed piecemeal microautophagy of the nucleus or PMN), which is induced via nitrogen starvation, the contact sites between the nuclear ER and the vacuole bulge into invaginations of the vacuolar membrane. Finally, a micronucleophagic vesicle surrounded by the vacuole membrane and containing some nuclear ER and non-essential parts of the nucleus buds off and is degraded within the vacuole lumen [10,11,12]. Atg15 further degrades MVB vesicles, and the involvement of Atg15 in lipolytic activity against neutral lipids and in the regulation of lipid droplet homeostasis has been reported [13,14,15], but this does not explain its putative direct role in membrane lysis. Remarkably, a later study found that Atg15 acted as a phospholipase with a substrate preference for phosphatidylserine (PS) but with lesser activity; also, cardiolipin and phosphatidylethanolamine were hydrolyzed [16]. During the course of this project, freeze–fracture electron microscopy showed the accumulation of PS on the outer membrane leaflet of autophagic bodies, while it was rare on the inside of the vacuole [17]. This would explain how Atg15 can mediate the lysis of autophagic bodies without affecting the integrity of the vacuole. However, how micronucleophagic vesicles surrounded by parts of the vacuolar membrane are lysed remains open. The lysis of intravacuolar ABs further requires the presence of vacuolar proteinase A and B [18]. This is surprising since cells deficient in proteinase B still possess significant vacuolar proteolytic activity. Also, the dependency of AB lysis on the vacuolar H^+^-ATPase is mechanistically not well understood [19].

Here, we found that Atg15 was not present at phagophores or ABs. For the lysis of ABs, Atg15 can be located at the vacuolar membrane or the lumen. The unexpected behavior of a putative cytosolic Atg15 variant prompted us to carefully dissect the Atg15 membrane topology, resulting in the identification of only a single transmembrane span. Remarkably, only the overexpression of membrane-embedded biologically active Atg15 variants affected cell growth. We found that these growth defects were independent of vacuolar proteinase A and vacuolar acidification.

## 2. Materials and Methods

### 2.1. Strains, Media, and Growth Conditions

The yeast strains listed in Table 1 were derived from the WT strain WCG4 MATα *his3-11,15 leu2-3,112 ura3* [20] and BY MATa; *his3∆1*, *leu2∆0*, *met15∆0*, *ura3∆0*. Deletions and insertions were generated using the method described previously [21,22]. BY knockout strains were obtained from EUROSCARF, Frankfurt, Germany.

Yeast strains were cultivated at 30 °C and 220 rpm in yeast extract–peptone–dextrose medium (YPD, 1% yeast extract, 2% peptone, 2% glucose, pH 5.5) or synthetic complete medium (CM, 0.67% yeast nitrogen base w/o amino acids, 2% glucose, pH 5.6, supplemented with 0.0117% of the appropriate amino acids for auxotrophic selection). Autophagy was induced via nitrogen starvation: yeast cells were incubated in synthetic defined medium without nitrogen (SD-N, 0.17% yeast nitrogen base without amino acids and ammonium sulfate, 2% glucose).

Differences in growth rate were analyzed using serial drop tests on plates with different carbon sources. Serial 10-fold dilutions of o/n yeast cultures were spotted on CM agar plates supplemented with either glucose or galactose and incubated at 30 °C for 3–6 days prior to documentation.

### 2.2. Plasmids

Plasmids are listed in Table 2. P_MET_-GFP-Atg15: The gene for ATG15 was amplified from chromosomal DNA bracketed by the restriction sites for BamHI and SalI and inserted into the MCS of pUG34.

P_ATG15_-GFP-Atg15: ATG15 was chromosomally tagged with GFP at the N-terminus, the whole construct containing endogenous promotor and terminator was then amplified with the restriction sites for SacII and AscI and inserted into the MCS of pRS313.

2xCitrine-Atg15: For the construction of 2xCitrine-Atg15, a PCR product containing yECitrine bracketed by the restriction sites for BamHI and SpeI, was generated using the plasmid pKT101 [23] as a template. After digestion, it was then inserted into P_Atg15_-GFP-Atg15, replacing the GFP. To enhance the signal brightness, a second yECitrine was inserted upstream of the first, using SalI and BamHI.

Mutations in Atg15 were generated with site-directed mutagenesis using Atg15-3xHA (previously described in [6] as a template and mutagenic primers to induce the exchange of N173 and N208 to alanine in Atg15^1glyc^-3xHA and N173, N202 and N208 to alanine in Atg15^no glyc^-3xHA.

Atg15∆TMD-3xHA: For the construction of Atg15∆TMD-3xHA in an overexpression vector, the insert was amplified using pUE7 ([6]: Atg15-3xHA) as a template and a forward primer binding downstream of the TMD. The PCR product containing ATG15^36–520^-3xHA was then inserted in the vector pYES2 after digestion with NotI and KpnI. To replace the transmembrane domain of Atg15 (amino acids 2–35) with the sorting sequence of either Pho8 ALP (amino acids 1–58) or CPY (amino acids 1–25), DNA fragments coding for the transmembrane domains bracketed by sequences homologue to the insertion site in ATG15∆TMD-3xHA were generated via PCR. The PCR products were inserted via homologous recombination into plasmid Atg15∆TMD-3xHA digested with PvuII and HindIII after both DNA fragments were transformed into *atg15∆* cells.

For the construction of Atg15-Suc2His4, the plasmid pR90 expressing Pmt1-Suc2His4 [24] was digested with XhoI and SacI, and an amplicon containing one of several truncated versions of Atg15 was inserted via homologous recombination, replacing Pmt1. The primers were chosen to generate versions of Atg15 truncated after amino acid D51, D104, R136, T210, T256, R301, L359, R416, R436 or D473 as well as at L520 for full-length Atg15.

**Table 2 cells-12-02056-t002:** Plasmids used.

Plasmid	Genotype	Reference
P_MET_-GFP-Atg15	pUG34 CEN P_MET17_-GFP-ATG15-T_CYC1_	This study
P_ATG15_-GFP-Atg15	pRS313 CEN P_Atg15_-GFP-ATG15-T_ATG15_	This study
GFP	pUG34 CEN P_MET17_-GFP-T_CYC1_	[22]
2xCitrine-Atg15	pRS313 CEN P_ATG15_-2xyECitrine-ATG15-T_Atg15_	This study
Atg15-3xHA	pRS316 CEN P_ATG15_-ATG15-3xHA	[6]
Atg15^no glyc^-3xHA	pRS316 CEN P_ATG15_-ATG15^N173/202/208A^-3xHA	This study
Atg15^1glyc^-3xHA	pRS316 CEN P_ATG15_-ATG15^N173/208A^-3xHA	This study
mCherry-Atg8	pRS316 CEN P_ATG8_-mCherry-ATG8-T_Atg8_	[25]
P_Atg15_-Atg15∆TMD-3xHA	pRS426 2µ P_ATG15_-ATG15∆TMD-3xHA-T_CYC1_	This study
P_MET17_-Atg15∆TMD-3xHA		
P_GAL1_-Atg15∆TMD-3xHA	pYES2 2µ P_GAL1_-ATG15∆TMD-3xHA-T_CYC1_	This study
ALP-Atg15∆TMD-3xHA	pYES2 2µ P_GAL1_-ALP(1-58)-ATG15∆TMD-3xHA-T_CYC1_	This study
CPY-Atg15∆TMD-3xHA	pYES2 2µ P_GAL1_-CPY(1-58)-ATG15∆TMD-3xHA-T_CYC1_	This study
Atg15-HA		
Sec63-RFP	pJK59 P_Sec63_-Sec63-RFP	This study
GFP-Atg18	pRS316 CEN P_ATG8_-GFP-ATG8-T_ATG8_	[26]
Pgk1-GFP	pRS316 CEN P_PGK1_-PGK1-GFP-T_ADH1_	[27]
Atg15(1-520)-Suc2His4	YEp352 2µ ATG15^1−520^-SUC2HIS4C	This study

### 2.3. Western Blot Analysis

Two OD_600_ units of yeast cells were pelleted (5000 rpm, 5 min) and incubated for 10 min on ice in 1 mL alkaline lysis buffer (0.28 M NaOH, 1.125% (*v/v*) β-mercaptoethanol). Proteins were precipitated with trichloroacetic acid (TCA) in a final concentration (f.c.) of 7.5% (*v/v*), pelleted (18,000× *g*, 10 min, 4 °C), washed twice with ice-cold acetone, dried and resuspended in 2× Lämmli buffer (117 mM Tris pH 8, 3.4% (*w/v*) SDS, 12% (*w/v*) glycerol, 0.004% (*w/v*) bromophenol blue, 0.016% (*w/v*) β-mercaptoethanol). The proteins were then analyzed using SDS-PAGE and Western blotting, with rabbit anti-ApeI antibodies (Eurogentec, Seraing, Belgium) in a dilution of 1:5000 and mouse anti-GFP monoclonal antibodies (Roche Diagnostics, Mannheim, Germany) in a dilution of 1:10,000. Goat anti-rabbit and goat anti-mouse monoclonal antibodies conjugated to horseradish peroxidase were used as second antibodies, respectively, and detected with either Pierce^TM^ ECL Plus Western Blotting Substrate (ThermoFisher Scientific, Schwerte, Germany) or Amersham^TM^ ECL Western Blotting Detection Reagents (GE Healthcare, Solingen, Germany).

### 2.4. Fluorescence Microscopy

Fluorescence microscopy was performed with a DeltaVision microscope (Olympus IX71, Applied Precision, Issaquah, WA, USA) equipped with the UPlanSApo ×100, 1.4 numerical aperture (NA), oil immersion objective, a CoolSNAPHQ2TM couple-charged device (CCD) camera and different filter sets specific for GFP and mCherry. Imaging occurred with a 100× objective and 2 × 2 binning. At least 20 focal planes along the *z*-axis with a distance of 0.2 μm were captured, and the resulting images were deconvolved using softWoRxTM (Applied Precision, Issaquah, WA, USA) and further processed with Fiji [28].

### 2.5. Deglycosylation Assay

Cell expressions of different C-terminally tagged Atg15 constructs were grown to stationary phase, approx. 60 OD_600_ units, were harvested (3000 rpm, RT, 5 min), washed twice with cold TBS (20 mM Tris/HCl pH 7.6, 200 mM NaCl) and resuspended in 400 µL lysis buffer (50 mM HEPES/KOH pH 7.5, 140 mM NaCl, 1 mM EDTA, 10% glycerol, 0.5% sodium desoxycholate, 2% Triton X-100, 0.1% [*w/v*] SDS, 1 mM PMSF, Complete proteinase inhibitor Mix [Roche]) and vortexed with 400 µL glass beads at 4 °C for 30 min. Cell debris was removed by centrifugation (1× at 500× *g*, 4 °C, 5 min; 2× at 13,200 rpm, 4 °C, 5 min). The supernatant was first incubated with 3 µL anti-HA antibody (0.2 mg/mL) for 3 h at 4 °C on a rotating wheel and then 1 h with 6 mg protein A-sepharose. The samples were centrifuged, washed twice with lysis buffer and once with washing buffer (50 mM KH_2_PO_4_ pH 5.5, 0.02% [*w/v*] SDS) and resuspended in washing buffer supplemented with 0.1 M β-mercaptoethanol. One half of each sample was incubated with 15 mU endoglycosidase H (Roche) for 1 h at 37 °C, then 6 µL of 6× SDS sample buffer was added, and the samples were analyzed using SDS-PAGE and Western blotting.

### 2.6. Membrane Topology

Membrane association assay: Cells expressing different Atg15 constructs were grown to stationary phase, and 60-90 OD_600_ units were harvested, resuspended in breaking buffer (50 mM Tris/HCl pH 7.5, 10 mM EDTA, 1 mM PMSF, complete proteinase inhibitor mix) and lysed with glass beads. Cell debris was removed via centrifugation (500× *g*, 5 min, 4 °C) and the supernatant was mixed 1:1 with either breaking buffer, 2 M potassium acetate, 0.2 M Na_2_CO_3_, 2.5 M urea or 2% Triton X-100 and incubated for 45 min at 4 °C. Each sample was centrifugated (100,000× *g*, 1 h) to separate membrane-bound proteins (pellet) from cytosolic proteins (supernatant). The pellet was resuspended in SDS sample buffer, while the proteins in the supernatant were precipitated with 50% (*v/v*) TCA, washed twice with acetone and resuspended in SDS sample buffer. The samples were then analyzed using SDS-PAGE and Western blotting.

Membrane topology: Truncated versions of Atg15 were C-terminally tagged with His4c and Suc2, a yeast invertase, and transformed into STY50 yeast cells. The cells were then streaked on CM plates lacking uracil and histidine but supplemented with 6 mM L-histidinol (CM -Ura -His +L-histidinol) and CM plates without uracil as a control (CM -Ura). Growth was analyzed after 3–6 days of incubation at 30 °C. The glycosylation pattern of the Atg15-His2Suc4 constructs was analyzed as described before, using 3 µL anti-invertase antibody (provided by Prof. Lehle, Regensburg, Germany).

### 2.7. Statistical Analyses

Blots were quantified using the free software Fiji 2.14.0 [28]. Statistical analyses for Western blots as well as fluorescence microscopy, were performed using GraphPad Prism 10.0.2 (GraphPad software, Boston, MA, USA). Graphs were plotted using the mean value together with the standard error of the mean (SEM). Statistical relevance was determined using the unpaired two-tailed *t*-test and is indicated in the graphs as follows: not significant (n.s. or no asterisk) for *p* > 0.05, * for *p* < 0.05, ** for *p* < 0.01, *** for *p* < 0.001 and **** for *p* < 0.0001.

## 3. Results

### 3.1. Atg15 Is Not Localized at the Phagophore or at Autophagic Bodies

Previous work showed that small amounts of Atg15 in the vacuole were sufficient for the lysis of autophagic bodies. We were thus interested if some Atg15 is present at autophagic bodies and could thus mediate the lysis of the membrane, where it is embedded. Vacuolar breakdown of autophagic cargos can be followed in western blots by detecting the proteolysis stable GFP [29]. We used a biologically active GFP-Atg15 and measured its vacuolar targeting during nitrogen starvation. Defects in the MVB-pathway in *vps4∆* cells led to an almost complete block of vacuolar transport and degradation of GFP-Atg15, while autophagy defects in *atg1∆* cells did not reduce its targeting; in contrast, a somewhat higher degradation occurred (Figure 1A,B). This suggests that most GFP-Atg15 is not targeted to the vacuole via autophagy but via the MVB pathway.

In line, fluorescence microscopy showed in the vacuole of *pep4∆* cells ABs positive for mCherry-Atg8, but within the detection limit, no GFP-Atg15 was colocalizing (Figure 1C).

To allow for the identification of small levels of Atg15 with high sensitivity, we generated a 2xCitrine-Atg15 construct with its endogenous promotor to avoid overexpression artifacts. In line with previous findings, 2xCitrine-Atg15 colocalized in growing cells with the ER-marker Sec63-RFP (Figure 2A). Interestingly, after 4 h starvation in nitrogen-free SD-N medium, we now could further clearly detect 2xCitrine-Atg15 at the vacuolar membrane (Figure 2A). Phagophores can be best visualized in fluorescence microscopy after overexpression of the selective cargo prApe1. This leads to the formation of a giant cargo [30], which is engulfed by the phagophore. Even after longer exposure times, no 2x-Citrine-Atg15 was detectable at the phagophores labeled with mCherry-Atg8 (Figure 2B). Together, these data suggest that Atg15 is not targeted to ABs and that the lysis of ABs is mediated by Atg15 at MVB-vesicles and/or the vacuolar membrane.

### 3.2. Generation and Characterization of Atg15-Chimeras

In a long-standing effort, we aimed to dissect the molecular function of Atg15 by generating chimeric variants relocated to either the vacuole lumen, the vacuolar membrane or the cytosol. During this project, a partially overlapping study was published [8]; we thus omit overlapping aspects and focus on additional novel findings. To redirect Atg15, the first step was to identify its targeting signal. We speculated that the sorting information of Atg15 to the MVB pathway may be within its amino-terminal transmembrane domain (TMD). Indeed, GFP-Atg15-(2-40)-Pep12 was redirected to the vacuolar lumen, indicating its entry into the MVB pathway. These results are not shown since Hirata et al. also showed that fusion of the Atg15-TMD to GFP is sufficient for its entry into the MVB pathway [8].

We used this to generate CPY-Atg15-∆TMD by replacing the transmembrane domain of Atg15 (amino acids 2–35) with the sorting signal (aa 1–25) of carboxypeptidase Y (CPY). We expected that this chimera would be targeted to the vacuolar lumen. Indeed, digestion with endoglycosidase H confirmed glycosylation of CPY-Atg15-∆TMD, indicating its entry into the ER to Golgi sorting pathway (Figure 3A). Indirect immunofluorescence of HA-tagged CPY-Atg15-∆TMD further demonstrated its localization in the vacuole and at the ER in *atg15∆ pep4∆* cells, while in *atg15∆* cells, it was only detected at the ER due to its vacuolar degradation (Figure 3B). Interestingly, CPY-Atg15∆TMD showed wild-type-like biological activity shown via the maturation of prApe1 (Figure 3C) and in the lysis of MVB vesicles followed by the degradation of the marker protein GFP-Cps1 (Figure 3D).

We additionally generated an GFP-ALP-(1-58)-Atg15∆TMD-(2-35) construct by replacing the Atg15 transmembrane domain with the sorting signal of the alkaline phosphatase Pho8. As expected, this construct reached the vacuolar membrane and was also biologically active. This is not shown due to the overlap with an GFP-ALP-(1-52)-Atg15-∆TMD-(37-520) construct presented in [8]. Together we found that Atg15 constructs are active when either localized at the vacuolar membrane or in the vacuolar lumen.

We next aimed to generate a soluble cytosolic Atg15 variant to analyze its activity, but also to check if it would be harmful for the cell. Since such a construct will not enter the ER to Golgi pathway, its glycosylation would be impaired. We thus further analyzed the relevance of glycosylation for Atg15 activity. Atg15 contains three predicted N-glycosylation sites at Asn173, Asn202 and Asn208. Mutation of all three potential glycosylation sites to alanine (Atg15 no glyc) reduced but did not block its activity (Figure 4A). The presence of the Asn202 glycosylation site alone was sufficient to restore the activity, showing that glycosylation at Asn202 alone is sufficient for Atg15 activity (Figure 4A). We thus generated an Atg15-∆TMD construct by truncating the Atg15 transmembrane domain (amino acids 2–35). Surprisingly, GFP-Atg15-∆TMD with the inducible *MET25* promotor showed almost half of the wild-type autophagic activity measured using prApe1 maturation, even when the promotor was not fully active in the presence of 0.3 mM methionine (Figure 4B); furthermore, the lysis of MVB vesicles was detectable (Figure 4C). Also, [8] reported partial activity of an *ADH1*-Atg15-∆TMD construct. Partial activity of Atg15-∆TMD was surprising; we therefore aimed to achieve further insights into this unexpected behavior. As shown previously, a block of the MVB pathway in *vps4∆* cells had only a slight effect on prApe1 maturation [5], while vacuolar targeting of GFP-Atg15 is severely affected (Figure 1), indicating that small amounts of Atg15 are sufficient for vesicle lysis. The activity of GFP-Atg15-∆TMD, however, was strongly abolished in *vps4∆* cells (Figure 4D,E), underlining the importance of the MVB pathway in this case. We further expressed Atg15-∆TMD under control of the endogenous *ATG15*-, the *MET25*- and the strong *GAL1*-promotor and measured the lysis of autophagic bodies via maturation of prApe1 and the lysis of the MVB-vesicles using the release of GFP from Cps1-GFP. With both selective cargos, we found a clear dosage dependence of the activity (Figure 4B,C)).

In fluorescence microscopy, GFP-Atg15-∆TMD, with its endogenous promotor, predominantly localized to a punctum close to the inside of the nuclear ER, marked with Sec63-RFP (Figure 4F), which is reminiscent of the INQ (intranuclear quality control compartment). The INQ is a protein deposit of misfolded proteins [31,32]. In contrast, [8] reported localization of *ADH1*-GFP-Atg15-∆TMD in the whole nucleus. Based on the observed Vps4 requirement of the Atg15-∆TMD activity, we speculated that this construct might still contain transmembrane domains, which might act as a sorting signal, targeting a small portion of the MVB pathway. Indeed, we found that Atg15-∆TMD does not behave as a soluble protein but appeared peripherally membrane-associated (Figure 4G,H). However, while both wild-type Atg15 and CPY-Atg15-∆TMD were glycosylated as shown by Endo H digestion, Atg15-∆TMD showed no glycosylation, not even when overexpressed with the strong *GAL1*-promotor (Figure 3A). This suggests that the majority of Atg15-∆TMD does not enter the ER and Golgi sorting pathway.

### 3.3. Experimental Dissection of the Membrane Topology of Atg15

Due to the unexpected activity of Atg15-∆TMD, we speculated that Atg15 might contain additional membrane domains. Indeed, in addition to the transmembrane domain near the amino terminus, several potential additional membrane domains might be present (Figure 5A). To experimentally dissect the membrane topology of Atg15, we used an elegant dual-tag approach [33,34]. In this approach, the protein of interest is truncated after the putative transmembrane spans, and a tag consisting of His4C and Suc2 is fused to its carboxy terminus. His4C is a fragment of the His4 protein and possesses histidinol dehydrogenase activity. When expressed in a *his4∆* yeast strain, His4C allows the strain to grow on L-histidinol media, when the His4C-tag is exposed to the cytosol. To allow for the uptake of L-histidinol into the cells, they additionally carry a mutation in the *HOL1* gene. On the other side, glycosylation of the Suc2 tag can be detected using Western blots when the tag is exposed to the lumen of the ER and Golgi. To avoid interference with endogenous Suc2, its gene is also deleted in the strain used. The advantage of this approach is that cytosolic localization of the dual tag is indicated by growth on L-histidinol, while luminal exposure results in significantly altered glycosylation (Figure 5C).

We generated 11 truncated Atg15 versions (Figure 5C, the numbers in the figure indicate at which amino acids Atg15 was truncated) and found that all were unable to grow on L-histidinol but showed strong glycosylation (Figure 5B,E). This shows that Atg15 has only a single transmembrane domain near its amino terminus. Thus, the membrane association is most likely due to hydrophobic regions. It is thus unclear why Atg15-∆TMD shows Vps4-dependent activity. Probably, some of the protein reaches the vacuole in MVB vesicles and is set free via spontaneous lysis.

### 3.4. Overexpression of Membrane-Embedded Atg15 Variants Affects Growth, Dependent on the Active Site Serine 332

We analyzed the effect of ALP-Atg15∆TMD, CPY-Atg15-∆TMD, and Atg15-∆TMD, all expressed under the control of the strong GAL1-promotor on cellular growth. On none-inducing glucose media, no growth differences were detectable. Remarkably, on galactose media, only the expression of the membrane-embedded variant ALP-Atg15-∆TMD and Atg15 itself, but neither CPY-Atg15-∆TMD nor Atg15-∆TMD, caused reduced growth (Figure 6A). The replacement of the lipase active site serine 332 of Atg15 with alanine (ALP-∆TMD-S332A) significantly restored growth, showing that the growth defect is linked to the enzymatic activity of Atg15 (Figure 6A). This interesting finding allowed us to monitor Atg15 activity also in cells defective for vacuolar proteinase A or the V-ATPase. So far, this has not been possible since both proteinase A and the V-ATPase are required for the lysis of intravacuolar vesicles [18,19]. The V-ATPase consists of a membrane-embedded V_o_ and a peripherally attached V_1_ part [35]. We analyzed cells lacking either a peripheral subunit of the V-ATPase or a component of the integral membrane Vo-part (Figure 6B). Again, the overexpression of active ALP-Atg15-∆TMD reduced growth, while the inactive ALP-Atg15-∆TMD-S332A led to better growth. This indicates that the observed growth defect depends on the Atg15 activity but is independent of the vacuolar pH (Figure 6B). Also, in atg15∆ pep4∆ cells lacking vacuolar proteinase A, ALP-Atg15-∆TMD overexpression affected growth, while the overexpression of ALP-Atg15-∆TMD-S332A showed better growth (Figure 6C). This shows that under overexpression conditions, Atg15 activity affects growth even in the absence of proteinase A.

To analyze in more detail how the overexpression of ALP-Atg15-∆TMD affects the growth rate of cells, we plated out aliquots and counted after 3 days the number of colonies. The colony number of atg15∆ cells was set to 100%, the number of colonies with inactive ALP-Atg15-∆TMD-S332A was 82.9% and those expressing the active ALP-Atg15-∆TMD 38.3% (Figure 6D). After 5 days, no increased number of colonies was observed, suggesting that ALP-Atg15-∆TMD affects the colony-forming ability of the cells.

## 4. Discussion

We were first interested if Atg15 only attacks the membrane in which it is embedded. If this holds true, Atg15 should be present in AB, at least in small parts, which might have escaped detection so far. However, mCherry-Atg8 positive Abs, which accumulated in *pep4∆* cells lacking vacuolar proteinase A, showed in fluorescence microcopy no colocalization with GFP-Atg15 (Figure 1). Additional sensitive analyses with 2xCitrine-Atg15 and a giant cargo generated via prApe1 overexpression allowed direct visualization of elongating phagophores, but again, no GFP-Atg15 was found at phagophores (Figure 2). Also, our CPY-Atg15-∆TMD construct lacking transmembrane domains was fully active (Figure 3). We conclude that for the lysis of ABs, Atg15 does not need to be at their membrane, and the localization of the Atg15 active site motif in the vacuolar lumen is sufficient. Atg15 can thus be either present at MVB vesicles, the vacuole membrane or within the vacuole lumen.

On the other side, the growth defect caused by overexpression of Atg15 but not of Atg15-S332A variants was restricted to membrane-embedded variants such as ALP-Atg15-∆TMD and to Atg15 itself, but not to CPY-Atg15-∆TMD (Figure 6). This growth defect is thus most likely generated by Atg15 activity directed against its embedding membrane. Atg15 activity against PS is significantly higher than those against PE, for example [16]. Thus, within the vacuolar lumen, Atg15 could preferentially hydrolyze PS present at the membrane outsides of ABs but absent from the inside of the vacuole membrane [17]. However, upon overexpression, the small activity of Atg15 against other lipids might be sufficient to be detrimental for cellular growth.

This growth defect allowed us to further evaluate the role of proteinase A and the V-ATPase. Indeed, overexpression of active membrane-embedded Atg15 variants affected the growth of *pep4∆* cells deficient in vacuolar proteinase A and of cells deficient either for a subunit of the V_o_ or the V_1_-part of the V-ATPase (Figure 6). This indicates that at least when overexpressed, Atg15 activity does not depend on proteinase A or vacuolar acidification. However, what could then be the role of proteinase A and vacuolar acidification in lysing ABs? The most obvious function would be the proteolytic activation of Atg15 in an acidic environment. However, we could never get hints for Atg15 maturation. Alternatively, proteinase A might be required to gain Atg15 access to the vesicle membrane. But it should also be considered that defective proteolysis in the vacuole might have an inhibitory feedback effect on vesicle lysis. Indeed, defects in amino acid efflux in *atg22∆* cells have been linked with delayed vesicle lysis and, thus, AB accumulation [36].

It should be emphasized that the growth defect for overexpressed Atg15, which we observed even in the absence of proteinase A and vacuolar acidification, does not completely rule out proteolytic activation of Atg15 within the vacuole. It is also well possible that on its sorting pathway to the vacuole, Atg15 has only a small intrinsic activity against its embedding membrane, which at its endogenous expression level is compensated and only acts as a membrane quality control. However, upon overexpression, such an activity might become harmful to the cell. In this scenario, Atg15 might then be fully activated inside the vacuole. In line, we observed growth defects dependent on the presence of the active site serine 332 when overexpressing ALP-Atg15-∆TMD in *vps11∆* cells. Vps11 is a component of the HOPS complex, which is involved in vacuole fusion events [37]. This could suggest membrane attack before Atg15 reaches the vacuole. Together our data indicate that overexpressed membrane-embedded Atg15 is active without proteinase A or vacuolar acidification.

Since Atg15-∆TMD was biologically active and dependent on the presence of Vps4 and still membrane-associated (Figure 4), we speculated that additional transmembrane domains might be present and mediate entry into the MVB-pathway. But our in-depth topological analyses of Atg15 (Figure 5) demonstrated that Atg15 has only a single transmembrane domain near its amino terminus and that the amino terminus is in the cytosol, while the carboxy terminus and the active site motif is exposed into the ER lumen. How Atg15-∆TMD reaches the Vps4-dependent vacuole remains open. It could either be taken up into the lumen of MVB-vesicles and set free by spontaneous lysis of some of these vesicles in the vacuole. Alternatively, the INQ-like deposit might be partially targeted Vps4-dependent to the vacuole for degradation, where Atg15-∆TMD can then act.

Based on our data, we speculated that Atg15 might just act as a phospholipase within the vacuole. We were thus curious if redirecting another yeast phospholipase to the vacuole might be able to substitute for Atg15. We choose phospholipase Plb2 due to its comparable substrate specificity [38]. We aimed to replace the Plb2 sorting sequence with the Atg15-TMD and truncated Plb2 at amino acid 679. We thus generated *MET25*::mCherry-Atg15-TMD-Plb2-(19-679) and *MET25*::mCherry-Atg15-TMD-Plb2-(27-679). Both constructs were detectable via fluorescence microscopy in the vacuole but were unable to complement the prApe1 maturation defect of *atg15∆* cells (not shown). Negative results are hard to interpret, so future work will be necessary to better understand a putative unique role of Atg15 in lysing bio-membranes.

## 5. Conclusions

Our experiments showed that Atg15 only has a single transmembrane domain near its amino terminus. We found its amino terminus to be exposed to the cytosol, while the rest of the protein, including the lipase active site motif, is within the ER lumen. We found no hints for targeting of Atg15 to the vacuole via autophagy. Atg15 chimeras targeted to either the vacuolar membrane or lumen were biologically active. Interestingly, overexpression of membrane-embedded Atg15 variants negatively affected growth independent of vacuolar acidification and vacuolar proteinase A. These growth defects were dependent on the presence of the Atg15 active site serine 332.

## Figures and Tables

**Figure 1 cells-12-02056-f001:**
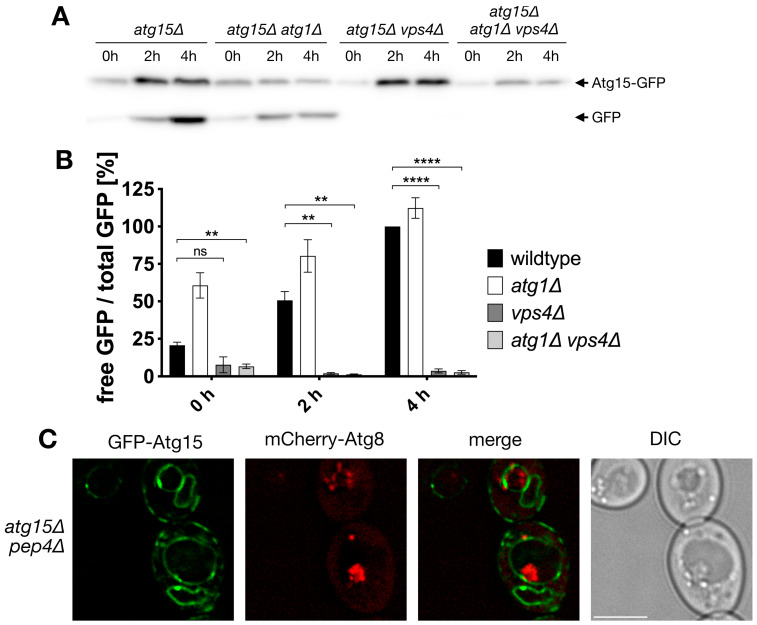
Atg15-GFP is not targeted to the vacuole via autophagy and is absent from autophagic bodies within the detection limit. (**A**) Vacuolar targeting and degradation of Atg15-GFP are followed in immunoblots with antibodies against GFP. Cells were incubated for the indicated times in nitrogen-free SD-(N) medium. (**B**) Quantification of (**A**), Statistical relevance: not significant (ns), ** for *p* < 0.01 and **** for *p* < 0.0001. n = 3. (**C**) Immunofluorescence microscopy of autophagic bodies marked with mCherry-Atg8. Bar: 5 µm.

**Figure 2 cells-12-02056-f002:**
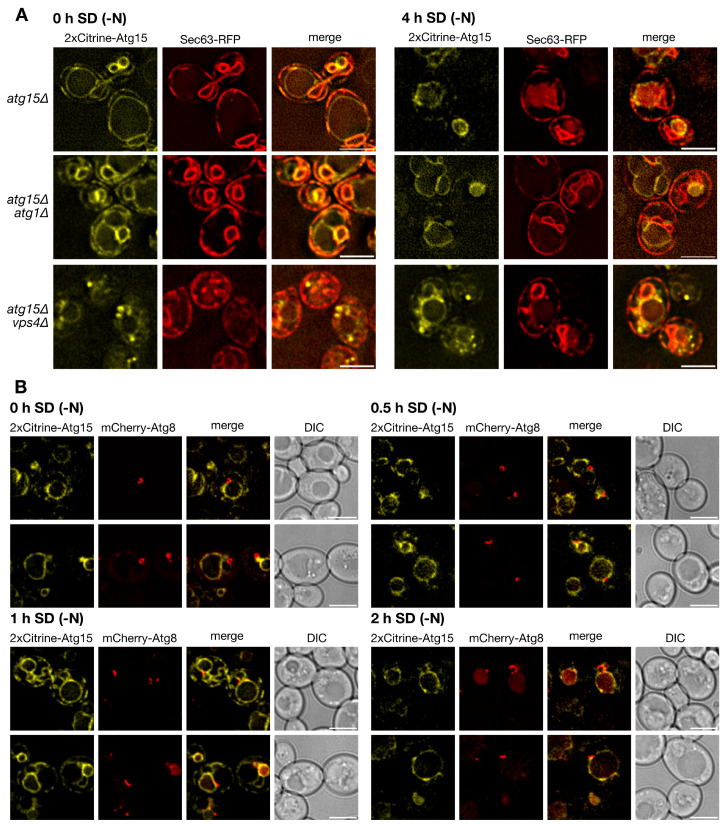
Localization of 2xCitrine Atg15. (**A**) 2xCitrine-Atg15 localizes to the ER marked with Sec63-RFP and after 4 h in SD-(N) to the vacuole membrane. (**B**) As detailed in the text, overexpression of prApe1 generates a giant cargo allowing direct visualization of the phagophore marked with mCherry-Atg8. No 2xCitrine-Atg15 was detectable at phagophores labeled with mCherry-Atg8. Bar: 5 µm.

**Figure 3 cells-12-02056-f003:**
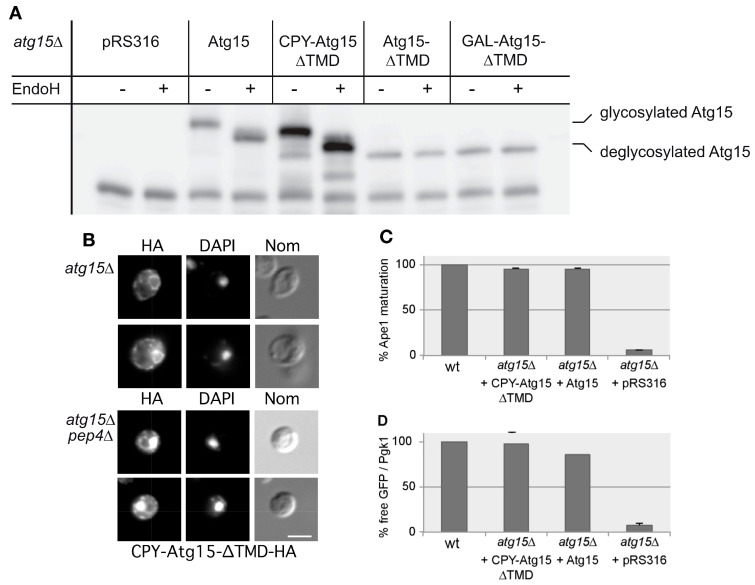
Characterization of Atg15 variants. (**A**) Endoglycosidase H digestions show glycosylation of CPY-Atg15-∆TMD, while Atg15-∆TMD is not glycosylated. (**B**) Indirect immunofluorescence microscopy using HA antibodies confirms vacuolar localization of CPY-Atg15-∆TMD-HA in *atg15∆ pep4∆* cells. DAPI indicates nuclear staining. Bar: 5 µm. (**C**,**D**) Biological activity of the Atg15 variants was monitored in Western blots of cells starved 4 h in SD-(N). Quantification of prApe1 maturation (n = 4) is shown in (**C**), and quantification of Cps1-GFP degradation (n = 4) in (**D**). In (**D**) the free GFP generated via the degradation of Cps1-GFP is normalized to the loading control Pgk1.

**Figure 4 cells-12-02056-f004:**
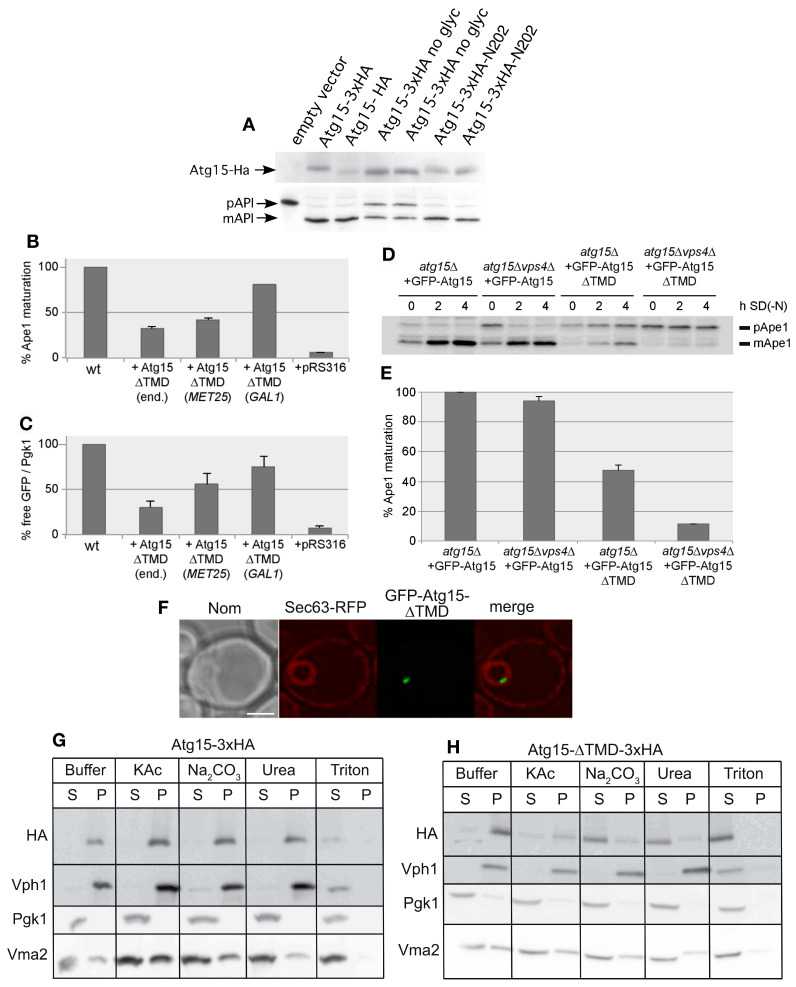
Characterization of Atg15-∆TMD. (**A**) Block of glycosylation of Atg15 at N173, N202 and N208 only partly affects the maturation of prApe1, restoration of glycosylation at N202 restores activity. (**B**,**C**) Biological activity of Atg15-∆TMD is dependent on the expression level in cells starved for 4 h in SD-(N) monitored by prApe1 maturation (n = 4) (**B**) or by Cps1-GFP degradation (n = 4), which is a marker for MVB-vesicle breakdown (**C**). (end.: endogenous promotor). For Cps1-GFP degradation, the free GFP level is normalized to the Pgk1 level, which acts as loading control. (**D**,**E**) Biological activity of Atg15-∆TMD followed by prApe1 maturation is Vps4 dependent. (**E**) Quantification of Western blots as shown in (**D**), n = 3. (**F**) Fluorescence microscopy shows the accumulation of GFP-Atg15-∆TMD in a punctate structure in the nucleus. ER is marked with Sec63-RFP. (**G**,**H**) Membrane association of Atg15 (**G**) and Atg15-∆TMD (**H**). Glass bead lysed cells were incubated with the indicated reagents and, via centrifugation at 100,000× *g*, separated into a supernatant (S) and pellet (P) fraction. As controls soluble Pgk1, peripherally membrane-associated Vma2 and integral membrane Vph1 were detected with respective antibodies.

**Figure 5 cells-12-02056-f005:**
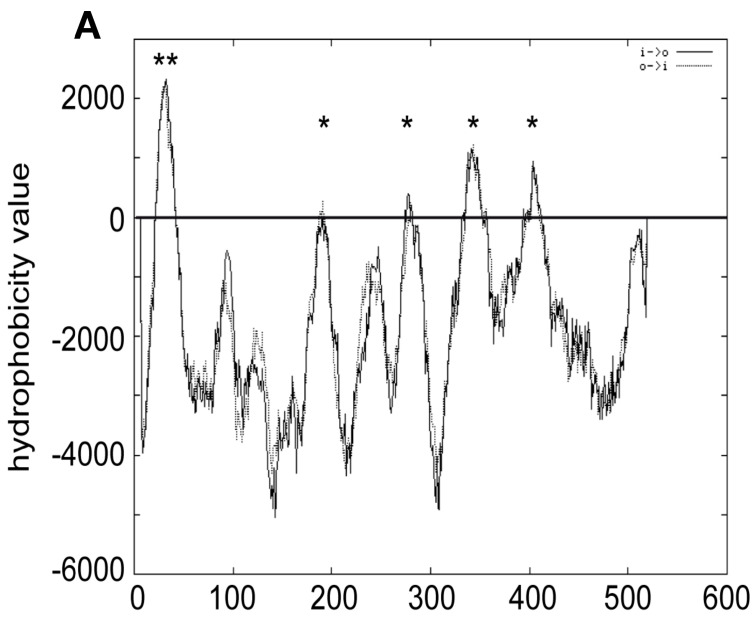
Membrane topology of Atg15. (**A**) Prediction of Atg15 transmembrane domains with TMPRED. ** indicates highly probable, * putative transmembrane domains. Atg15 constructs were truncated at the amino acids shown in (**C**) and fused with a tag containing both His4 and Suc2 (**D**). Exposure of this dual tag to the cytosol restores growth on plates containing histidinol (**B**). The colonies on plates are arranged as shown in (**C**). Exposure of the tag to the ER lumen results in glycosylation, probed by endoglycosidase H digestion (**E**).

**Figure 6 cells-12-02056-f006:**
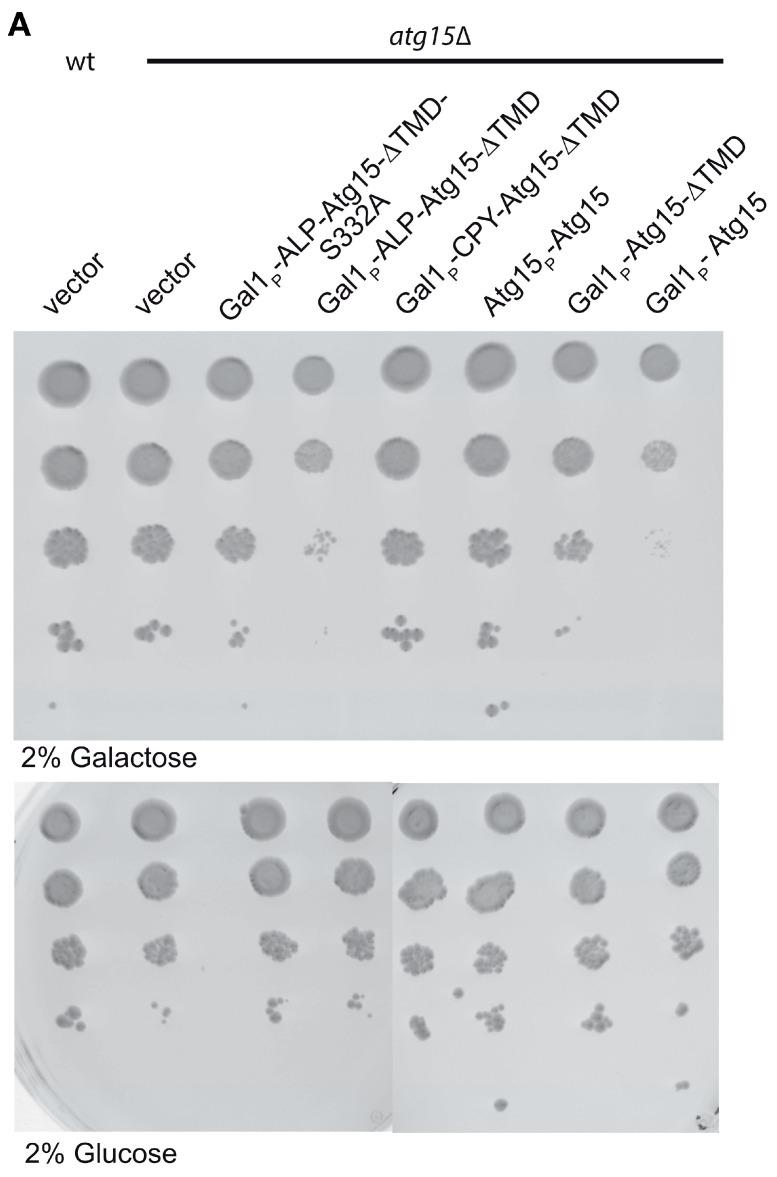
Overexpression of membrane-embedded Atg15 variants affects growth dependent on the active site serine 332. (**A**) 10-fold dilution series of liquid cultures were dropped on CM plates containing either glucose or galactose to induce the *GAL1*-promotor of the indicated constructs. (**B**) Growth defects are also detectable in mutants lacking subunits of the V-ATPase. The cartoon of the V-ATPase is based on [35]. (**C**) Growth defects in cells lacking the vacuolar proteinase A (*pep4∆*) are seen in cells expressing ALP-Atg15∆TMD but not endogenous Atg15. (**D**) Aliquots of *atg15∆* cells expressing the indicated constructs were plated out after 3 days in galactose medium, and the number of colonies formed was counted. SEM of 4 experiments is shown.

**Table 1 cells-12-02056-t001:** Strains used.

Strain	Genotype	Reference
WCG *atg15∆*	*atg15∆*::KAN	[6]
WCG *atg15∆ atg1∆*	*atg15∆*::KAN *atg1∆*::KAN	[6]
WCG *atg15∆ vps4∆*	*atg15∆*::KAN *vps4∆*::KAN	This study
WCG *atg15∆ atg1∆ vps4∆*	*atg15∆*::KAN *atg1∆*::natNT2 *vps4∆*::KAN	This study
WCG *atg15∆ pep4∆*	*atg15∆*::KAN *pep4∆*::KAN	This study
WCG *atg15∆ pep4∆ vps4∆*	*atg15∆*::KAN *pep4∆*::hphNT1 *vps4∆*::KAN	This study
WCG *atg15∆atg1∆ pep4∆ vps4∆*	*atg15∆*::KAN *atg1∆*::natNT2 *pep4∆*::hphNT1 *vps4∆*::KAN	This study

## Data Availability

Data sharing is not applicable to this article.

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
