# Peer review of "Topology and Function of the S. cerevisiae Autophagy Protein Atg15"

_cells, 2023, doi:10.3390/cells12162056_

Round 1

Reviewer 1 Report

General Impression

The authors describe the outcomes of a series of experiments designed to elucidate the trafficking, activity and topology of a key enzyme of yeast autophagy, the lipase Atg15. Written on the background of another recently published article on Atg15 function, the present paper aims to provide novel details to the big picture of membrane autophagy which still remains somewhat unclear. The manuscript summarizes a large body of work and utilizes broad spectrum of well-established techniques of yeast molecular biology, biochemistry and cell biology. The review of the theoretical background is sound and includes the discussion of recent overlapping data from a competing group. The experimental strategy is well thought-out, executed and documented.  

The authors provide evidence for the fact that Atg15 is not targeted to the vacuole via autophagy but instead resides in MVB and the ER. Western blotting and immunofluorescence studies on Atg15 variants show that the enzyme is active without its single N-terminal transmembrane domain and that the activity depends on glycosylation of a single critical asparagine residue. Finally, spot growth assays show negative effects of overexpression of membrane-associated variants of Atg15 that is independent of vacuolar acidification and proteinases. The discussion accurately summarizes the presented data and provides an appropriate interpretation of the findings:  There is still a lot to speculate about Atg15 function, and the manuscript does not add much clarity.

Points of criticism

Title: While the experiments in this paper describe several new aspects of Atg15 function, they do not actually show membrane lysis. Please consider changing to a more accurate description.

Figure 3A: The use of the +/- symbols for Endo H treatments is somewhat confusing. Might they have been switched?

There are few minor issues with the language that could be fixed by a copy editor

Author Response

Points of criticism

Title: While the experiments in this paper describe several new aspects of Atg15 function, they do not actually show membrane lysis. Please consider changing to a more accurate description.

As suggested, we changed the title to: "Topology and function of the S. cerevisiae autophagy protein Atg15".

Figure 3A: The use of the +/- symbols for Endo H treatments is somewhat confusing. Might they have been switched?

Unfortunately, the +/- symbols were indeed switched, we corrected this mistake in the novel Figure 3A.

Reviewer 2 Report

Very nice work. I found your manuscript to be scientifically sound and pleasant to read.  I have several suggestions for improvement:

1-sentence in lines 225-227 should be rewritten.

2-reference is made to prApe1 visualization by fluorescence microscopy (lines 233-234) but prApe1 was not included in Figure 2A or B. Perhaps indicate as "data not shown".

3- lines 247-248, regarding your reference to an overlapping study, can you cite the paper?

4-lines 283-286 contain two somewhat redundant sentences. Why is the second stated as "not shown"?

5-line 424, inclusion of a table with the cell counting data would be helpful.

The quality of the English is good with only minor edits required in some sentences. I will leave the editorial analysis up to a specialist.

Author Response

Comments and Suggestions for Authors

Very nice work. I found your manuscript to be scientifically sound and pleasant to read.  I have several suggestions for improvement:

1-sentence in lines 225-227 should be rewritten.

We rewrote the sentence to: " In line, fluorescence microscopy showed in the vacuole of pep4∆ cells ABs positive for mCherry-Atg8, but within the detection limit no GFP-Atg15 was colocalizing (Fig.1C)."

2-reference is made to prApe1 visualization by fluorescence microscopy (lines 233-234) but prApe1 was not included in Figure 2A or B. Perhaps indicate as "data not shown".

In Figure 2A,B we overexpressed non-labeled prApe1, which results in formation of a giant cargo and then detected the phagophore with mCherry-Atg8. To clarify this we rewrote the sentence to " Phagophores can be best visualized in fluorescence microscopy after overexpression of the selective cargo prApe1. This leads to the formation of a giant cargo [30], which is engulfed by the phagophore."

3- lines 247-248, regarding your reference to an overlapping study, can you cite the paper?

We inserted the reference Hirata et al 2021 MBoC, as suggested.

4-lines 283-286 contain two somewhat redundant sentences. Why is the second stated as "not shown"?

We agree that the second sentence is somewhat redundant and thus deleted it to improve the readability.

5-line 424, inclusion of a table with the cell counting data would be helpful.

As suggested, we changed the presentation of the cell counting data and found a bar diagram to be most intuitive. We included this as novel Figure 6D.

Reviewer 3 Report

This article provides important mechanistic information about the function of ATG15. 

Major points: 

The authors need to specify the number of experiments and whether western blots or imaging data is representative of 3 independent experiments. 3 independent experiments are required for key experiments. Especially when quanitifcations are shown, n numbers and the way the quantification was performed needs to be clearly stated in the figure legends. 

Author Response

This article provides important mechanistic information about the function of ATG15. 

Major points: 

The authors need to specify the number of experiments and whether western blots or imaging data is representative of 3 independent experiments. 3 independent experiments are required for key experiments. Especially when quanitifcations are shown, n numbers and the way the quantification was performed needs to be clearly stated in the figure legends. 

As suggested, we explain in Material and Methods the quantification and statistical analyses. Furthermore, we inserted the n numbers of all quantifications in the figure legends. All quantifications are based on at least three independent experiments.

Round 2

Reviewer 3 Report

My comments have been addressed.